# Finger-Based Numerical Training Increases Sensorimotor Activation for Arithmetic in Children—An fNIRS Study

**DOI:** 10.3390/brainsci12050637

**Published:** 2022-05-12

**Authors:** Christina Artemenko, Silke Maria Wortha, Thomas Dresler, Mirjam Frey, Roberta Barrocas, Hans-Christoph Nuerk, Korbinian Moeller

**Affiliations:** 1Department of Psychology, University of Tuebingen, 72076 Tuebingen, Germany; hc.nuerk@uni-tuebingen.de; 2LEAD Graduate School & Research Network, University of Tuebingen, 72072 Tuebingen, Germany; thomas.dresler@med.uni-tuebingen.de (T.D.); k.moeller@lboro.ac.uk (K.M.); 3Department of Neurology, University Medicine Greifswald, 17475 Greifswald, Germany; silkemaria.wortha@med.uni-greifswald.de; 4Department of Psychiatry and Psychotherapy, Tuebingen Center for Mental Health, University Hospital of Tuebingen, 72076 Tuebingen, Germany; 5Department Clinical Psychology & Experimental Psychopathology, University of Groningen, 9712 TS Groningen, The Netherlands; m.i.frey@rug.nl; 6Leibniz-Institut für Wissensmedien, 72076 Tuebingen, Germany; r.barrocas@iwm-tuebingen.de; 7Centre for Mathematical Cognition, School of Science, Loughborough University, Loughborough LE11 3TU, UK; 8Individual Development and Adaptive Education Center, 60323 Frankfurt am Main, Germany

**Keywords:** finger counting, finger-based training, arithmetic, embodiment, subbase-5 effect, single-digit addition, fNIRS

## Abstract

Most children use their fingers when learning to count and calculate. These sensorimotor experiences were argued to underlie reported behavioral associations of finger gnosis and counting with mathematical skills. On the neural level, associations were assumed to originate from overlapping neural representations of fingers and numbers. This study explored whether finger-based training in children would lead to specific neural activation in the sensorimotor cortex, associated with finger movements, as well as the parietal cortex, associated with number processing, during mental arithmetic. Following finger-based training during the first year of school, trained children showed finger-related arithmetic effects accompanied by activation in the sensorimotor cortex potentially associated with implicit finger movements. This indicates embodied finger-based numerical representations after training. Results for differences in neural activation between trained children and a control group in the IPS were less conclusive. This study provides the first evidence for training-induced sensorimotor plasticity in brain development potentially driven by the explicit use of fingers for initial arithmetic, supporting an embodied perspective on the representation of numbers.

## 1. Introduction

Children typically use their fingers when learning to count, understand numerical magnitudes, and calculate (e.g., [1]). This close association of fingers and numbers during early numerical development is reflected in finger-based strategies observed for counting, magnitude understanding including part-whole-relations, as well as arithmetic (for reviews see [2,3,4]).

As regards the working mechanism for finger-based numerical representations, it may not be accidental that our ten fingers reflect a base of 10, on which the place-value structure of the Arabic number system is built. The subdivision into five fingers on each hand adds a structural subbase-5 to finger-based numerical representations which allows to specifically evaluate finger-based numerical representations, particularly in the context of arithmetic performance. For instance, Domahs et al. [5] observed that children committed more specific arithmetic errors that deviated from the correct result by ±5—and thus by one full hand—than would be expected based on the split between the correct result and the erroneous solution. Moreover, even adults were found to show a subbase-5 effect in mental arithmetic, suggesting significantly increased difficulty in arithmetic problems that cross the subbase-5 boundary and thus require calculations to move from one to the other hand (e.g., 4 + 3 vs. 6 + 1) [6]. Even tasks as simple as magnitude comparison still seem to be influenced by the transparency of the subbase-5 structure of finger counting in a given culture in adulthood [7]. Taken together, finger-related strategies for counting and calculation in childhood seem to facilitate the development of a finger-based embodied representation of numbers, traces of which are still present in adults (see also [2]).

Within the framework of embodied cognition [8], finger-based representations of numbers are argued to be based on sensorimotor experiences when using fingers for counting and calculating (e.g., [9,10]). In line with this theoretical idea, behavioral findings on the association between fingers and numbers are complemented by the observation of an overlap in neural structures associated with finger movements and number processing (e.g., [11,12,13,14]). For instance, Tschentscher et al. [15] observed that the mere presentation of small numbers (1–5) led to increased activation in sensorimotor areas in the hemisphere controlling the hand commonly used by participants to start counting on their fingers [15]. Since this sensorimotor activation was found in the absence of any overt finger movements, the processing of these small numbers seemed to specifically coactivate sensorimotor areas associated with finger movements on the hand participants would normally use to count to the respective numbers on their own fingers. It is important to note in this respect that while there is a certain flexibility in counting procedures, the preferred starting hand is quite stable across time [16]. This suggests that the perception of numbers associated with one’s preferred counting habit seems to activate lateralized finger-based sensorimotor representations. However, most of the research on this topic is correlational and thus does not allow for causal inferences on whether such associations have a functional role. Furthermore, the findings reported above were restricted to the somatosensory cortex. Nevertheless, a neural overlap for fingers and numbers might also be detected in neural areas primarily associated with number processing, when number magnitudes need to be processed and not just detected.

Generally, numbers are processed in a fronto-parietal neural network, including parietal brain regions such as the intraparietal sulcus (IPS), which is associated with number magnitude and arithmetic processing, the posterior superior parietal lobule (PSPL), associated with attentional processes on the mental number line, and the angular gyrus (AG) and supramarginal gyrus (SMG), associated with arithmetic fact retrieval (for a model and its extensions see [17,18]). Interestingly, the neural correlates of mental arithmetic were observed to overlap with those of finger discrimination in the left IPS and bilateral PSPL [12]. Moreover, the neural correlates of number magnitude processing and hand grasping movements were found in adjacent areas in the bilateral IPS [19]. Comparable to these findings in adults, children also showed a neural overlap within the IPS for single-digit addition and finger movements [13], and more generally, right parietal activation during the comparison of the number of fingers presented on images of hands [20]. In sum, these studies suggest a critical role not only of the sensorimotor cortex, but also of the IPS for the association of fingers and numbers.

These findings from neuroimaging studies are further supported by first causal evidence on a neural association of fingers and numbers from studies with invasive electrical brain stimulation in patients [21] and transcranial magnetic stimulation (TMS) in healthy adults. The inhibition of the left motor cortex associated with hand movements by TMS induced deficits in counting [11]. The inhibition of parietal brain regions (left angular gyrus, supramarginal gyrus, regions close to IPS) by TMS or electrical brain stimulation induced deficits in both finger recognition and number processing ranging from number magnitude comparison to two-digit addition [14,21].

In the present study, we set out to investigate neural associations of fingers with number processing in children following a finger-based intervention. We pursued the idea that if children are intensively trained on using their fingers systematically for counting and calculations, this should facilitate the development of finger-based numerical representations. The finger-based intervention trained the systematic use of fingers for counting and initial calculation over the course of first grade [22]. We evaluated the neural correlates of mental arithmetic after this finger-based intervention in children, with a specific focus on activation in the sensorimotor cortex and the IPS, because these areas—but not frontal areas [12,23]—have been previously found to be associated with finger-based numerical representations. Specifically, we explored whether trained children differed in their sensorimotor and IPS activation for single-digit arithmetic from control children who did not complete the training by focusing on the subbase-5 effect in single-digit addition. To assess brain activation after the intervention, functional near-infrared spectroscopy (fNIRS) was used, since it tolerates movements, allows for an upright body position, can be applied in natural settings, and thus is appropriate for children [24].

## 2. Materials and Methods

### 2.1. Participants

A total of 46 children were recruited from the experimental and control groups of a large-scale finger-based intervention study [22], in which children were trained in finger-based numerical representations (i.e., systematic use of fingers for counting and initial calculation) during their math instruction in first grade. All participants were right-handed (except for two children), with no history of neurological or mental disorders, and without a disease that influences brain metabolism. Two children did not participate in the study due to technical problems (*n* = 1) or withdrawal from participation (*n* = 1). The intervention group (*N* = 22; 12 boys, 10 girls; age: *M* = 7;6 years, *SD* = 4.0 months, *Range* = [6;11–8;4] years) was assessed at the end of grade 1 after the intervention, while the control group (*N* = 22; 9 boys, 13 girls; age: *M* = 7;10 years, *SD* = 5.8 months, *Range* = [7;1–8;10] years) was assessed seven months later due to organizational constraints, when children already attended second grade. (Please note that information on age is missing for two children in the control group; they attended the same grade and thus have approximately a similar age.)

Written informed consent was provided by the children’s parents or legal guardians, and oral assent was obtained from all children prior to testing. The study was approved by the local ethics committee of the Medical Faculty of the University of Tuebingen (620/2012BO2).

### 2.2. Finger-Based Intervention

Based on the model of Roesch and Moeller [3], (i) basic numerical skills (e.g., exact number word sequence combined with basic counting), (ii) quantity-number concept (e.g., finger pattern reflects specific cardinality of the counted set), and (iii) number relations (e.g., number composition, comparison, calculation) were trained. Additionally, finger gnosis was strengthened. The finger-based training included different tasks and mini-games, whereby a special focus was to instruct children to use their fingers in a consistent and systematic way. The finger-based intervention was targeted at number relations, including addition and subtraction procedures. A total of 18 training sessions of about 30 min each were administered by the experimenters over the duration of the first school year during regular school lessons for trained children. Control children attended the regular math curriculum in school without a special focus on the use of fingers. In the intervention study, all children in a classroom were assigned to either the training or the control group in a quasi-experimental manner based on the willingness of teachers to participate in the finger-based training.

Before the intervention, the intervention and control group did not significantly differ in finger gnosis, *t*(40) = −0.59, *p* = 0.561. Furthermore, the groups did not significantly differ in their early numerical abilities as assessed by a standardized math test administered at the beginning of first grade (i.e., four subtests on *ordinality*, *symbolic–nonsymbolic mapping*, *nonsymbolic set comparison*, and *symbolic magnitude comparison* of the ERT 0+, Eggenberger Rechentest 0+ [25]), *F*(1, 37) = 2.83, *p* = 0.101. Additionally, there were no significant group differences in custom built tasks on addition, *F*(1, 37) = 2.45, *p* = 0.126, subtraction, *F*(1, 37) = 0.004, *p* = 0.949, number line estimation from 0 to 10, *F*(1, 37) = 0.038, *p* = 0.846, or number line estimation from 0 to 20, *F*(1, 37) < 0.001, *p* = 0.988. All comparisons of numerical abilities controlled for influences of general cognitive abilities (i.e., two subtests on *continuing rows* and *matrices* of the CFT 1-R, Culture Fair Intelligence Test–Revised [26]). Note that for some children pretest data was missing (*n* = 2 for finger gnosis and *n* = 4 for numerical and/or general cognitive abilities).

### 2.3. Experimental Task

All computerized tasks were programmed in Presentation (NeuroBehavioral Systems, Inc., Berkeley, CA, USA). The stimuli were presented in the center of the screen using white font against a black background (font size of 100). Instructions emphasized both speed and accuracy. Responses were noted by the experimenter and reaction times (RT) were recorded by voice key or button press.

The addition task consisted of 40 single-digit addition problems. Each item consisted of two single-digit operands with a single-digit result ranging from 3 to 9. Half of the items did not require a subbase-5 carry operation (e.g., 3 + 1; 6 + 2), so that they could be solved within the same hand from a finger counting perspective. The other half did require a subbase-5 carry operation (e.g., 4 + 3; 1 + 8), so that their solution would require crossing from one hand to the other when applying a finger-based approach. The numerical size of the result (problem size) was matched between conditions. Numbers 5 and 10 were not used as either operands or results [6]. In an event-related design (see Figure 1A), the stimuli were presented in Arabic notation in random order for 4 s, followed by an inter-stimulus-interval of 4–7 s (jittered in steps of 0.150 s, mean of 5.5 s). Participants solved the addition problems by mental calculation and responded orally by saying “is [number]”.

### 2.4. Procedure

After the finger-based intervention, trained and control children were tested individually in a dimly lit separate room in their school during regular school hours. At the beginning, instructions for all tasks and two practice items per task were provided. During the fNIRS measurement, three numerical tasks related to finger counting were conducted: the addition task, the number partner task, and a (non-) symbolic number magnitude comparison task. The focus of the present study is on the addition task because it was conducted with an event-related design leading to a better signal-to-noise ratio as compared to the other tasks with a block design. The results of the number partner task are reported in the Appendix A. The number comparison was not analyzed, since it was too difficult for the children. Task order was counterbalanced across participants. All measurements were jointly performed by two experimenters, one responsible for monitoring the fNIRS measurement and the other for instructing participants and documenting their responses.

### 2.5. fNIRS Data Acquisition

The fNIRS data was measured using the continuous wave ETG-4000 Optical Topography System (Hitachi Medical Corporation, Tokyo, Japan). This system uses wavelengths of 695 ± 20 nm and 830 ± 20 nm as light sources and continuously measures data at a sampling rate of 10 Hz. The optodes (16 sources, 14 detectors) were arranged in two 3 × 5 arrays with a fixed inter-optode distance of 30 mm covering the left and right hemisphere, respectively, embedded in a cap (Brain Products GmbH, Herrsching, Germany). The probesets were localized at P3/P4 for the upper channels in the back and oriented towards F3/F4 for this channel row according to the 10–20 system (see Figure 1B); for more details on the location about the probeset, see [27]. The correspondence of fNIRS channels to the underlying cortical areas was estimated based on a virtual registration method [28,29,30].

### 2.6. Data Analysis

*Data exclusion*. In the addition task, participants were excluded due to drop out from all analysis (*n* = 1), and due to missing data from behavioral data analysis (*n* = 2) or RT analysis only (*n* = 2). For the RT analysis, RTs of incorrectly solved trials, missing RTs, RTs below 200 ms, and RTs deviating more than three median absolute deviations from the participant’s median were excluded. The same trials were excluded from the fNIRS analysis since the addition task followed an event-related design.

*fNIRS data preprocessing*. fNIRS data were analyzed with custom MATLAB (The MathWorks, Inc., Natick, MA, USA) scripts. Relative concentration changes of oxygenated (O_2_Hb) and deoxygenated hemoglobin (HHb) were calculated for each fNIRS channel according to the modified Beer-Lambert law. The fNIRS signal was preprocessed by using the temporal derivative distribution repair (TDDR) [31] to correct for high-amplitude motion artifacts and by applying a bandpass filter of 0.01–0.5 Hz. The correlation-based signal improvement (CBSI) [32] was used to reduce low-amplitude motion artifacts; CBSI is based on the negative correlation between O_2_Hb and HHb and considered as one of the best artifact correction methods [33]. CBSI was necessary since the signal of the relatively young children in this study was quite noisy due to movement artifacts. The remaining noisy channels were interpolated by surrounding channels (13.22% in the addition task). In the addition task, trials that were excluded from the RT analysis (13.60%) as well as trials containing uncorrectable artifacts (1.89%) were further excluded from analysis. The event-related fNIRS data of the addition task was analyzed in a model-based approach. Based on the standard hemodynamic response function, a general linear model was computed for each channel, participant, and condition.

Two regions of interest (ROIs) were defined for either hemisphere: the sensorimotor cortex and the areas around the IPS (see Figure 1B; cf. Figure 1 in [27]). The sensorimotor cortex (consisting of the primary somatosensory cortex, primary motor cortex, pre-motor cortex, and supplementary motor cortex) was chosen to reflect the finger localizer in the study of Tschentscher et al. [15] and is represented by the channel 16 on the left [coordinates: −59 −20 49] and 38 on the right [coordinates: 16 −21 49]. The areas around the IPS (adjacent to the inferior and superior parietal lobule) were chosen to reflect the observed overlap of parietal activation for finger discrimination and arithmetic in the study of Andres et al. ([12], Figure 2D) and is represented by the channel 19 on the left [coordinates: −42 −56 59] and 44 on the right [coordinates: 42 −57 60].

*Statistical data analysis.* Statistical analyses were conducted using JASP (Jeffreys’s Amazing Statistics Program, Version 0.13.1, JASP Team, 2016). The behavioral data of the addition task was analyzed using ANOVAs discerning the factors group (trained vs. control children) and carry 5 (without vs. with) for both RT and accuracy. For the fNIRS data, ANOVAs were conducted for each ROI (sensorimotor cortex and IPS) discerning the factors group (trained vs. control children), carry 5 (without vs. with), and hemisphere (left vs. right). Post-hoc tests were added when necessary. As this study was exploratory with open hypotheses, we did not apply statistical corrections for multiple comparisons. Nevertheless, we focused on only two ROIs rather than evaluating brain activation through all channels of the fNIRS probeset. Violin and bar plots were created using the R package ggplot2 [34].

## 3. Results

### 3.1. Behavioral Data

The RT analysis revealed a significant main effect of carry 5, *F*(1, 37) = 16.28, *p* < 0.001, $\eta_{p}^{2}$ = 0.306, that was qualified by a significant interaction of group and carry 5, *F*(1, 37) = 8.67, *p* = 0.006, $\eta_{p}^{2}$ = 0.190 (see Figure 2). The subbase-5 effect, reflecting that the children solved addition problems without carry 5 faster than addition problems with carry 5, was more pronounced in trained children than control children, *t*(37) = 2.94, *p* = 0.006, *d* = 0.94. A simple effects analysis revealed a significant subbase-5 effect only for trained children (279 ms; *p* < 0.001; see Figure 2), but no subbase-5 effect for control children (44 ms, *p* = 0.282). The main effect of group was not significant, *F*(1, 37) = 0.554, *p* = 0.461, $\eta_{p}^{2}$ = 0.015.

The analysis of accuracy revealed a significant main effect of carry 5, *F*(1, 39) = 9.90, *p* = 0.003, $\eta_{p}^{2}$ = 0.202, indicating that children solved addition problems without carry 5 more accurately than addition problems with carry 5. Neither the main effect of group, *F*(1, 39) = 0.433, *p* = 0.514, $\eta_{p}^{2}$ = 0.011, nor the interaction of group and carry 5 were significant, *F*(1, 39) = 2.05, *p* = 0.161, $\eta_{p}^{2}$ = 0.050.

### 3.2. fNIRS Data

For the *sensorimotor cortex*, the main effect of carry 5, *F*(1, 41) = 5.38, *p* = 0.025, $\eta_{p}^{2}$ = 0.116 was qualified by the significant three-way interaction of group, carry 5, and hemisphere, *F*(1, 41) = 7.06, *p* = 0.011, $\eta_{p}^{2}$ = 0.147. Breaking down this three-way interaction into its constituting two-way interactions revealed a different pattern for trained and control children: In trained children, a marginal significant main effect of carry 5 was observed, *F*(1, 21) = 4.21, *p* = 0.053, $\eta_{p}^{2}$ = 0.167, but no main effect of hemisphere, *F*(1, 21) = 0.98, *p* = 0.334, $\eta_{p}^{2}$ = 0.044, and no interaction, *F*(1, 21) = 1.47, *p* = 0.239, $\eta_{p}^{2}$ = 0.065. In control children, a significant interaction of carry 5 and hemisphere, *F*(1, 20) = 6.39, *p* = 0.020, $\eta_{p}^{2}$ = 0.242, was observed, while the main effects for carry 5, *F*(1, 20) = 1.34, *p* = 0.259, $\eta_{p}^{2}$ = 0.063, and hemisphere, *F*(1, 20) = 2.32, *p* = 0.143, $\eta_{p}^{2}$ = 0.104, were not significant. A subsequent simple effects analysis further indicated that trained children showed a subbase-5 effect with higher activation for addition problems with carry 5 than without carry 5 in the right sensorimotor cortex, *t*(21) = 2.18, *p* = 0.041, *d* = 0.456, but not in the left sensorimotor cortex, *t*(21) = 1.50, *p* = 0.148, while control children showed a subbase-5 effect with higher activation for addition problems with carry 5 than without carry 5 in the left sensorimotor cortex, *t*(21) = 2.58, *p* = 0.018, *d* = 0.564, but not in the right sensorimotor cortex, *t*(21) = −0.32, *p* = 0.752 (see Figure 3B). No significant effects were observed for group, *F*(1, 41) = 0.371, *p* = 0.546, $\eta_{p}^{2}$ = 0.009, or hemisphere, *F*(1, 41) = 3.14, *p* = 0.084, $\eta_{p}^{2}$ = 0.071, or for the interaction between group and carry 5, *F*(1, 41) = 0.865, *p* = 0.358, $\eta_{p}^{2}$ = 0.021, group and hemisphere, *F*(1, 41) = 0.138, *p* = 0.713, $\eta_{p}^{2}$ = 0.003, or carry 5 and hemisphere, *F*(1, 41) = 0.935, *p* = 0.339, $\eta_{p}^{2}$ = 0.022.

For the *IPS*, only the interaction of group, carry 5, and hemisphere was significant, *F*(1, 41) = 4.59, *p* = 0.038, $\eta_{p}^{2}$ = 0.101. Breaking down this three-way interaction into its constituting two-way interactions revealed no significant effects, neither for trained children nor for control children (note also that a different separation by hemisphere did not explain the three-way interaction). The interaction might indicate a stronger subbase-5 effect in the left IPS in trained children, however, the comparison to control children was not significant, *t*(42) = 1.57, *p* = 0.124 (see Figure 3C). No significant main effects were found for group, *F*(1, 41) = 0.677, *p* = 0.416, $\eta_{p}^{2}$ = 0.016, carry 5, *F*(1, 41) = 0.258, *p* = 0.614, $\eta_{p}^{2}$ = 0.006, or hemisphere, *F*(1, 41) = 2.44, *p* = 0.126, $\eta_{p}^{2}$ = 0.056. Additionally, the interactions between group and carry 5, *F*(1, 41) = 0.827, *p* = 0.369, $\eta_{p}^{2}$ = 0.020, group and hemisphere, *F*(1, 41) = 1.17, *p* = 0.287, $\eta_{p}^{2}$ = 0.028, and carry 5 and hemisphere, *F*(1, 41) = 0.038, *p* = 0.847, $\eta_{p}^{2}$ < 0.001, were not significant.

## 4. Discussion

This study set off to explore the neural signature of finger-based numerical representations by comparing neural activation in the sensorimotor cortex and IPS during arithmetic between children who completed a finger-based intervention and a control group. Differences due to the finger-based intervention were found in the sensorimotor cortex with a different lateralization of the subbase-5 effect in mental addition. Importantly, this was observed in the absence of overt finger movements. Activation differences in the IPS between trained and control children were less conclusive.

Activation associated with the subbase-5 effect showed a different lateralization in the sensorimotor cortex of children after the finger-based intervention as compared to control children. While the subbase-5 effect was more pronounced in the right sensorimotor cortex in trained children, it was also found in the left sensorimotor cortex in control children. The right-lateralized activation in the sensorimotor cortex, which is typically associated with finger and hand movements, was observed during addition in trained children and thus seems to be an intervention effect: Over the course of their first school year, children were trained in using their fingers for counting and initial arithmetic with their dominant hand for small numbers up to five and with both hands for larger numbers. Since most children participating in the study were right-handed, they would have started with their right hand and used their left hand when crossing the 5 boundary (for evidence that free counting mostly starts with the preferred hand in children see [35], in adults see [36]). In turn, this may have led to more contralateral (right-hemispheric) activation in the sensorimotor cortex when contrasting addition with vs. without crossing the 5 boundary (for contralateral activation depending on the hand to start finger counting in adults see also [15]). While all addition problems that require subbase-5 carries have results above 5 (i.e., using the second hand additionally to the first hand, which is the left hand in right starters), addition problems that do not require subbase-5 carries equally consist of both problems calculated within the second hand (e.g., 6 + 3) and problems calculated within the first hand (e.g., 1 + 3). Although this is necessary to avoid confounds of carry with problem size, it may add some noise to the data as concerns activation of the second hand in finger-number representations: The second hand may be activated in 100% of the addition problems with crossing the subbase-5 boundary, but also in 50% of the other addition problems.

Nevertheless, we found more pronounced right sensorimotor activation in the absence of overt finger movements during single-digit arithmetic for trained children as compared to control children. We suggest that this might reflect traces of the finger-based intervention, which was designed to strengthen the association between fingers and numbers. The significant effects on lateralization, depending on whether the left or right hand is predominantly involved in the respective finger movements, was observed for the sensorimotor cortex (not for the parietal cortex), which is in line with previous findings in adults [12,15]. As such, these data allow for the generalization of such lateralization signatures to children who were trained to use their fingers for counting and initial calculations. In sum, the observed activation pattern in the sensorimotor cortex during mental arithmetic seems to reflect strengthened embodied numerical representations after a finger-based intervention. Whether or not these embodied representations facilitate arithmetic processing is a question for future research.

Besides finger-related sensorimotor activation, a neural overlap for fingers and numbers was previously also reported for the IPS in adults [12] and children [13]. In the current study, however, differences in IPS activation between trained and control children need to be interpreted with caution. Trained children tended to show a stronger subbase-5 effect in the left IPS, but the direct comparison to control children was not significant. This finding might reflect a representation of a finger-related effect in mental arithmetic in a typical brain region for number magnitude processing. This finger-related numerical representation might be established in children who were trained to use their fingers for single-digit arithmetic, comparable to what was previously found for adults [12]. Children at this early age begin to establish this representation leading to a subbase-5 effect in both solution times and accuracy (as in adults, cf. [6]); whether an explicit finger-based numerical training facilitates this development in the core region of number processing – the IPS – could not be clarified in the current study due to insufficient power. Future studies would be desirable to further evaluate the neural overlap of fingers and numbers in the IPS in longitudinal or intervention studies in children involving different age groups.

When interpreting the results of the current study, there are limitations to be considered. First, sample sizes in the present study were comparably small and the statistical power to detect small effects was accordingly low. Thus, we were careful in interpreting the absence of effects that might be detected with sufficient power. Note that the small sample sizes were caused by the special populations recruited (in particular an experimental and a control group of children from an intensive intervention study running over an entire school year, cf. [22]), meaning that the present study is nonetheless the largest neurocognitive intervention study investigating finger-based numerical representations in children to date. Second, constraints on the time available for reliable fNIRS measurements in such young children further resulted in low numbers of trials per task. This can affect the reliability of the data. Third, data quality was not optimal, with missing behavioral data and low signal-to-noise ratio in the fNIRS data, potentially due to the measurements in a natural setting at children’s schools. This was primarily due to motion artifacts, as children of this age may hardly sit still over an extended period of time and as some of the paradigms required movements. Although fNIRS can tolerate motion artifacts to a certain extent, motion artifact correction without the use of short channels was not optimal for this sample and setting. Fourth, concerning the control-group design, it needs to be taken into account that the neural activation was only assessed after the finger-based intervention took place (and not in a to-be-preferred pre-post design). Because children differ considerably in their numerical development at that age, this leads to additional error variance within groups. In the absence of general performance differences, the specificity of group differences in the subbase 5 effect seems to reflect a training effect rather than rather than being due to general differences between groups independent of the training. Supporting this interpretation, the trained and control groups of the larger intervention study did not significantly differ in numerical precursor abilities, general cognitive abilities and finger gnosis before the intervention. Moreover, due to constraints on study planning (ensuring testing of the trained children directly after the training, which was close to summer holidays) and logistics (bringing the fNIRS device to schools), the testing of the control group was delayed so that the groups differed in about half a year of age and education. Although differences in sensorimotor activation might not simply be explained by age differences, they are a serious confound, especially for the inconclusive results on IPS activation, because finger-based training and development might lead to similar activation changes in the IPS. Nevertheless, we are confident that the present data are meaningful because the children did not differ in their behavioral performance. Additionally, Krinzinger et al. [13] did not find age-related changes in the neural overlap of areas associated with finger movements and mental arithmetic in an even larger age range (6–12 years) than in the current study. Finally, methodological limitations also need to be mentioned. Even though fNIRS theoretically can reach activation in areas around the IPS, the specific location of the overlap of numbers and fingers in the IPS might not be covered (for further explanations see [37]).

In conclusion, we found converging neural evidence for the strengthening of finger-based numerical representations following a finger-based intervention in children in their first year of school. In particular, single-digit mental arithmetic elicited activation in the sensorimotor cortex associated with using the respective hand for solving the task—even in the absence of overt finger movements (cf. [15]). This suggests enhanced embodied representations following sensorimotor experiences made in the training. However, whether finger-based instruction also leads to a more pronounced neural association of fingers and numbers in the IPS needs to be clarified in future research. In conclusion, the present results indicate that training finger-based numerical strategies, i.e., the use of fingers for counting, base-10 representations, and initial calculations, seems to leave learning traces in the brain. As such, the current exploratory study may—despite its limitations—well serve as a starting point for better controlled future (intervention) studies substantiating the beneficial role of finger-based numerical representations for children’s numerical development.

## Figures and Tables

**Figure 1 brainsci-12-00637-f001:**
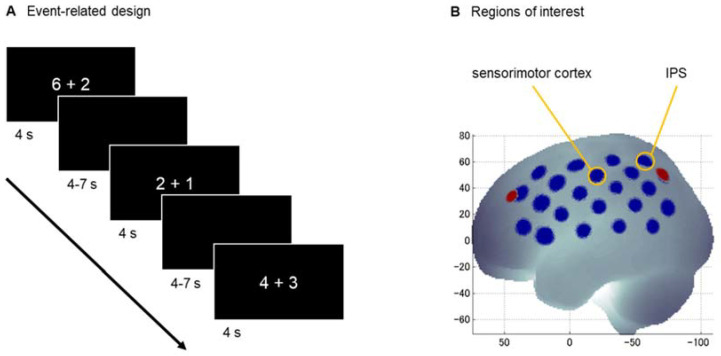
Experimental setup. (**A**) The addition task was presented in an event-related design and the children were asked to respond orally after mental calculation. (**B**) The channels (blue) for the regions of interest (yellow) are mapped on the cortical surface (same positions on the left and right hemisphere). The reference points are P3/P4 and F3/F4 (red) according to the 10–20 system. Abbreviation: IPS—intraparietal sulcus.

**Figure 2 brainsci-12-00637-f002:**
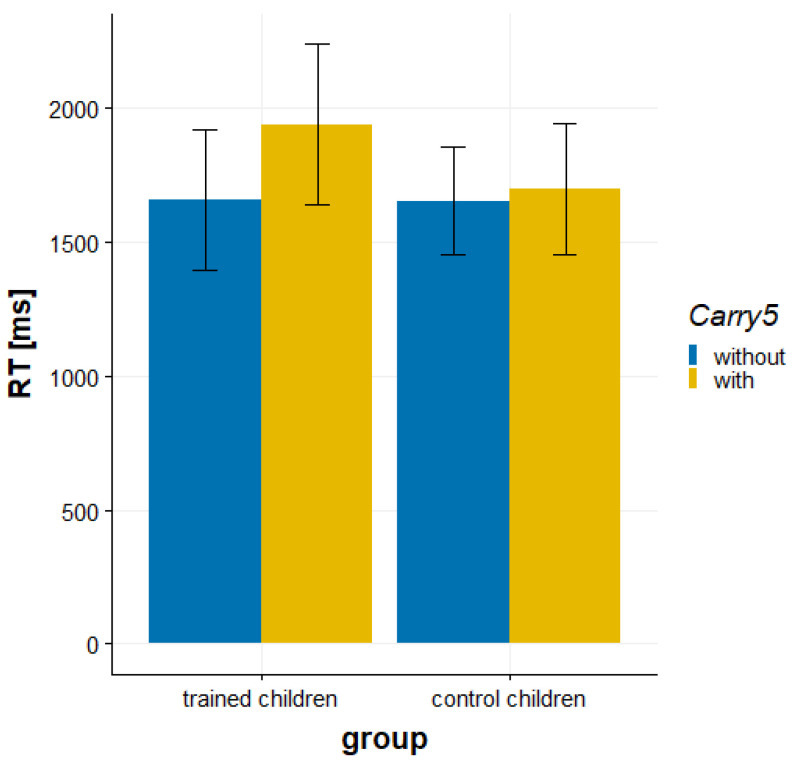
Behavioral data of the addition task. Trained children but not control children showed a subbase-5 effect in RT. Error bars indicate 95% CI.

**Figure 3 brainsci-12-00637-f003:**
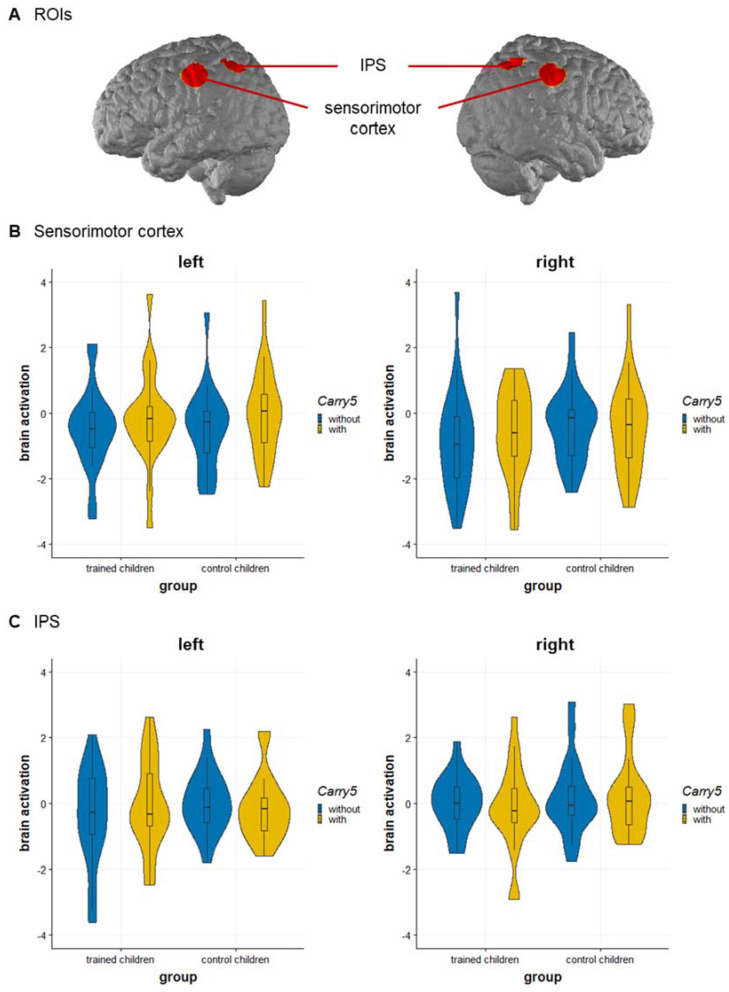
Neural activation of the sub-base effect in single-digit addition. (**A**) ROIs: Sensorimotor cortex and IPS on the left and right hemisphere. (**B**) Sensorimotor cortex: Trained children showed a subbase-5 effect in the right sensorimotor cortex and control children in the left sensorimotor cortex. (**C**) IPS: IPS activation for trained and control children for the subbase-5 effect. Violin plots display the kernel distribution of the neural activation within each ROI; box plots include the interquartile range (IQR = 25–75%) with the horizontal line representing the *Median*. Abbreviation: IPS—intraparietal sulcus.

## Data Availability

The data presented in this study are available on request from the corresponding author. The data are not publicly available due to ethical restrictions.

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
