# Peer review of "Finger-Based Numerical Training Increases Sensorimotor Activation for Arithmetic in Children—An fNIRS Study"

_brainsci, 2022, doi:10.3390/brainsci12050637_

Round 1

Reviewer 1 Report

The neural consequences of teaching calculation exclusively by finger counting in first grade were analyzed by fNIRS, in comparison with a control group.

Firstly I was surprised that the format of the article does not conform to the MDPI format. But this problem can be easily remedied.

I am comfortable with the conclusion of the authors, that results clearly indicate that training finger-based numerical strategies, i.e., the use of fingers for counting, base-10 representations and initial calculations, seem to leave learning traces in the brain. However, as my later quotation from Luria will show, it is to be hoped that these traces will rapidly become inoperative.

I am uncomfortable with the many flaws in the article. I list them rather in order of appearance than in order of importance.

  1. The authors never explain the acronym fNIRS: this should be done in the text when it is first used.
  2. The finger-based intervention, which it would be important to know, is not really described: reference is simply made to an unpublished document from the University of Tuebingen (Germany). I find it hard to believe that teachers can teach 6/7-year-olds that to calculate, for example 7 + 1, you can count the fingers of one hand by raising them (1, 2, 3, 4, 5), then continue with the other hand (6, 7), add another finger because you want to add 1 to the 7, and count the whole raised fingers again (1, ..., 8), to finally find the result 8!
  3. Students in the control group were tested 7 months after the students in the intervention group: as a result, the former are significantly older than the latter.
  4. The research has been approved by local ethics committee of the Medical Faculty of the University of Tuebingen. However, one may be concerned about the long-term consequences of the teaching practiced: do the members of this committee know, for example, that more than half a century ago the neuropsychologist Luria warned us that if a disabled student has been in a special class for a long time and continues to operate using external enumeration, he or she can certainly be classified as mentally retarded?
  5. The authors use violin diagrams but do not explain what these diagrams show. As this type of diagram is not (yet?) frequently used, this would be useful.
  6. Furthermore, Hadley (2016) is not in the references. The reference should probably be Wickham (2016), but the latter is also not listed in the references.
  7. I find it implausible that a sensory-motor area represents such an abstract concept as number. To me, it is merely an activation associated with learning that simply illustrates Hebb's old rule, summarized as "Cells that fire together wire together."
  8. Even the IPS, which the authors have not really demonstrated to retain traces of finger-based learning, is not well suited to a symbolic (largely verbal) representation of numbers.

Reviewer 2 Report

The aim of the current study was to measure the effect of a finger-based numerical training on arithmetic skills and brain activation, in children. The topic of the study in timely and interesting. However, I have some doubts on the validity of the conclusions draw from the results.  My major worrying is linked to the baseline condition. From my understand (but I could have missed this point) of the training procedure, there is no a baseline assessment (before training) of both behavioural math skills and brain activation for both trained and undrained children. If this is the case, I cannot see how the obtained results could be attributed to the effect of the training and not reflecting a difference between groups already present after the training. Without looking at the delta change (on both behavioural and brain activation) between pre and post training, I am not able to provide a definitive evaluation of the current study. When this (crucial) point will be clarified by the authors, I will be happy to revise the ms again.

Detailed comments:

p. 3. “The intervention group (N = 22; 12 boys, 10 girls; age: M = 7;6 years, SD = 4.0 months, Range = [6;11–8;4] years) was assessed at the end of grade 1 after the intervention, while the control group (N = 22; 9 boys, 13 girls; age1: M = 7;10 years, SD = 5.8 months, Range = [7;1–8;10] years) was assessed 7 months later due to organizational constraints, when children al- ready attended second grade.”

  • The training procedure is not described. I cannot figure out its timeline, which tasks were performed (and repeated) before and after the training as well as the non-finger training performed by the control group.

p. 4. “During the fNIRS measurement, three numerical tasks related to finger counting were conducted: the addition task, the number partner task, and a (non-)symbolic number magnitude comparison task.”

  • Only the addition task is described.

p. 4. “The focus of the present study is on the addition task because it had best data quality. As there were issues with task pro-cedures that led to reduced specificity of the neural data, the results of the number partner task are reported in the Supplementary Material.”

  • What is meant by data quality is not clear to me. If the author detected unreliable results, why should be then reported in the supplementary materials? Which would be their scientific utility for the readers?
  • What about the (non-)symbolic number magnitude comparison task? It is mentioned in the ms but is not described in the methods as well as in the results (not even in the supplementary materials).

p.6 Behavioral data

  • If I understand correctly (but I am probably wrong), the arithmetic task was administered only after the training. If this is the case (as there is no pre-training assessment) how can the effect of the training be extracted from the results? How can the causal interpretation of the results be justified? In other words, could the results shown in fig 2 (and those not shown for accuracy) simply reflect the characteristics of the two samples, already present after the training?

p.7 fNIRS data

  • I see here the same problem. If the fNIRS measurement has been only carried out after the training how can the effect of the training be extracted? If the Behavioral data cannot be interpreted as a change induced by the training, how the brain activation data can be then interpreted?
  • Where p-values corrected for multiple comparison?

Round 2

Reviewer 1 Report

I reread the article. I only saw one small (probable) error: in the lines 173-174 “two experiments” should be “two experimenters”.

On the substance, even if I do not always share the authors' method of approach or the educational goal they have in mind, I can agree with their cautious conclusion, “that training finger-based numerical strategies seem to leave learning traces in the brain”.

Reviewer 2 Report

I am still quite sceptical about whether this paradigm can reveal a causal role of training (and I am not convinced why the p-values were not correct) but the manuscript has certainly improved and the authors have done a good job in answering my questions. For these reasons I believe the manuscript, (despite the limitations) can have a useful function as a starting point for future (more focused) studies.
